# Development and Assessment of Seasonal Rainfall Forecasting Models for the Bani and the Senegal Basins by Identifying the Best Predictive Teleconnection

Luis Balcázar [1,2] , Khalidou M. Bâ [1,*] , Carlos Díaz-Delgado [1] , Miguel A. Gómez-Albores [1] , Gabriel Gaona [3,4] and Saula Minga-León [1]

1   Instituto Interamericano de Tecnología y Ciencias del Agua (IITCA), Universidad Autónoma del Estado de México, Toluca 50200, Mexico
2   Dirección de Posgrado, Universidad Técnica de Cotopaxi, Av. Simón Rodríguez s/n Barrio El Ejido Sector San Felipe, Latacunga 050108, Ecuador
3   Departamento de Recursos Hídricos y Ciencias Ambientales, Universidad de Cuenca, Cuenca 010203, Ecuador
4   Institute for Landscape Ecology and Resources Management (ILR), Justus Liebig University Giessen, 35390 Giessen, Germany
*   Correspondence: khalidou@uaemex.mx; Tel.: +722-29655-50/51 (ext. 111)

**Abstract:** The high variability of rainfall in the Sahel region causes droughts and floods that affect millions of people every year. Several rainfall forecasting models have been proposed, but the results still need to be improved. In this study, linear, polynomial, and exponential models are developed to forecast rainfall in the Bani and Senegal River basins. All three models use Atlantic sea surface temperature (SST). A fourth algorithm using stepwise regression was also developed for the precipitation estimates over these two basins. The stepwise regression algorithm uses SST with covariates, mean sea level pressure (MSLP), relative humidity (RHUM), and five El Niño indices. The explanatory variables SST, RHUM, and MSLP were selected based on principal component analysis (PCA) and cluster analysis to find the homogeneous region of the Atlantic with the greatest predictive ability. PERSIANN-CDR rainfall data were used as the dependent variable. Models were developed for each pixel of $0.25° \times 0.25°$ spatial resolution. The second-order polynomial model with a lag of about 11 months outperforms all other models and explains 87% of the variance in precipitation over the two watersheds. Nash–Sutcliffe efficiency (NSE) values were between 0.751 and 0.926 for the Bani River basin and from 0.175 to 0.915 for the Senegal River basin, for which the lowest values are found in the driest area (Sahara). Results showed that the North Atlantic SST shows a more robust teleconnection with precipitation dynamics in both basins.

**Keywords:** model; Sahel; SST; PERSIANN-CDR; RHUM; MSLP

## 1. Introduction

The Sahel is a semi-desertic region covering Africa's territory from the Atlantic Ocean to the Red Sea and dividing the Sahara Desert from the moist savannah. This region is widely known for being vulnerable to desertification, for its scarce water availability and rapid environmental degradation [1]. Indeed, interannual rainfall has seen important changes in the last five decades. Severe drought in the 1970s and 1980s brought famine and humanitarian crisis in the region [2–4]. While at the break of the current century, Samimi et al. [5] observed intense rainfall in 2007, equivalent to values with a return period of 1200 years. In addition, Biasutti [6] reports an increase in rainfall in the central and eastern Sahel, as well as a decrease in rainfall in the western Sahel, with intense and isolated rainfall.

Rainfall variability at the Sahel is dynamically related to the variability of atmospheric circulation, Hadley cells, and West African Monsoon (WAM) circulation [6]. WAM is a

coupled atmosphere-ocean-land system [7] responsible for summer rainfall in the Sahel from May to October [4,6], with the most significant rainfall happening in July–August–September (JAS) [8,9]. The WAM flows as a shallow moist surface air layer from the Gulf of Guinea, overlaid by the main northeast trade winds, which blow from the Sahara and which is known as Harmattan. The convergence of the trade winds and WAM form the Intertropical Convergence Zone (ITCZ), which is characterized by low pressure, laden with heat and moisture [10].

Sufficient and reliable observations are necessary to understand the great interannual variability of precipitation in the Sahel, and therefore the evolution of the hydrological regimes. However, in the Sahel, surface and high-altitude observations are very rare and when they do exist, their reliability is always questionable [8].

Weather forecasts are a result of field observation and general circulation models (GCM). However, due to the small amount of observed data and inconveniences of the MCG scale, it is not possible to satisfactorily answer key questions about the interrelation of atmosphere-ocean-land [11,12]. As a result, statistical models are more popular for applications that require a high spatial and temporal resolution scale [13]. Nonetheless, every climate model has future forecast uncertainty due to common systematic bias [6].

In West Africa, rainfall forecasting is performed by the Prévisions Climatiques Saisonnières en Afrique Soudano-Sahélienne forum (PRESASS, [14,15], formerly known as Prévisions Climatiques Saisonnières en Afrique de l'Ouest (PRESAO). Every year, between April and May, the forum is carried out to elaborate on that year's seasonal forecast. The event is summoned by the African Center of Meteorological Application for Development (ACMAD) and the CRA (Centre Regional de Formation et d'Application en Agrométórologie et Hydrologie Opérationelle—AGRHYMET) [16]. For the rainfall forecast, Climate Predictability Tool (CPT) and techniques such as statistical methods, dynamic models, and experts' judgment are used [13]. Forecasts are then compared to climate outlooks in international climate centers, and a consensus is reached to communicate the forecast to users. Such outlook is categorical, meaning that it consists of qualitative descriptions such as: above normal, normal, and below normal rainfall probability [13,16]. However, currently used PRESASS forum's forecasts and models are not precise enough, and drought and flooding catch authorities and people off guard. That is why it is necessary to develop better models [12,13].

The purpose of this study is to establish a forecasting model for rainfall during the rainy season (May–October) across the Bani and Senegal River basins in West Africa to propose appropriate insight for decision-making processes regarding water management. In order to achieve this objective, ocean-atmospheric variables and linear and non-linear models were used. Statistical techniques such as principal component analysis (PCA) and cluster analysis over Atlantic SST, RHUM, and MSLP variables were used while creating potential forecasting models to find the region with the highest rainfall predictive power. In linear, polynomial, and exponential models, SST was used as the only predicting factor. While in the linear stepwise regression model RHUM, MSLP covariables, Niño1 + 2, Niño3.4, Niño4, Oceanic Niño (ONI), and trans-Niño (TNI) indices were used, aside from SST. As a response variable for models, Precipitation Estimation from Remotely Sensed Information using Artificial Neural Networks—Climate Data Record (PERSIANN-CDR) data were used [17]. It is worth mentioning that forecasting was performed with the same spatial resolution of PERSIANN-CDR ($0.25° \times 0.25°$) for the 725 pixels that make up the Bani and the Senegal River basins.

## 2. Materials and Methods

### 2.1. Description of the Study Region

Sahel is the vast semi-arid region of Africa separating the Sahara Desert to the north from tropical savannas. From west to east, the Sahel stretches from northern Senegal—southern Mauritania to Eritrea and northern Ethiopia. This place is home of nearly 130 million people. The main means of livelihood are stockbreeding, fishing, and subsis-

tence agriculture. The latter is the most important sector and the main means of livelihood for most of the people who inhabit this region [18].

This study considers only the western part of the Sahel located between 13°W to 4°W and 8°N to 20°N, comprising the areas of the Bani River basin at Beneny Kegny hydrometric gauge (upper Niger) and the Senegal River basin at Bakel (Figure 1). Most of the basins' area is located in the Sahel (semi-arid climate), while the southern part of the basins has a tropical savanna climate, and the north of the Senegal River basin (center of Mauritania) corresponds to warm desert climate.

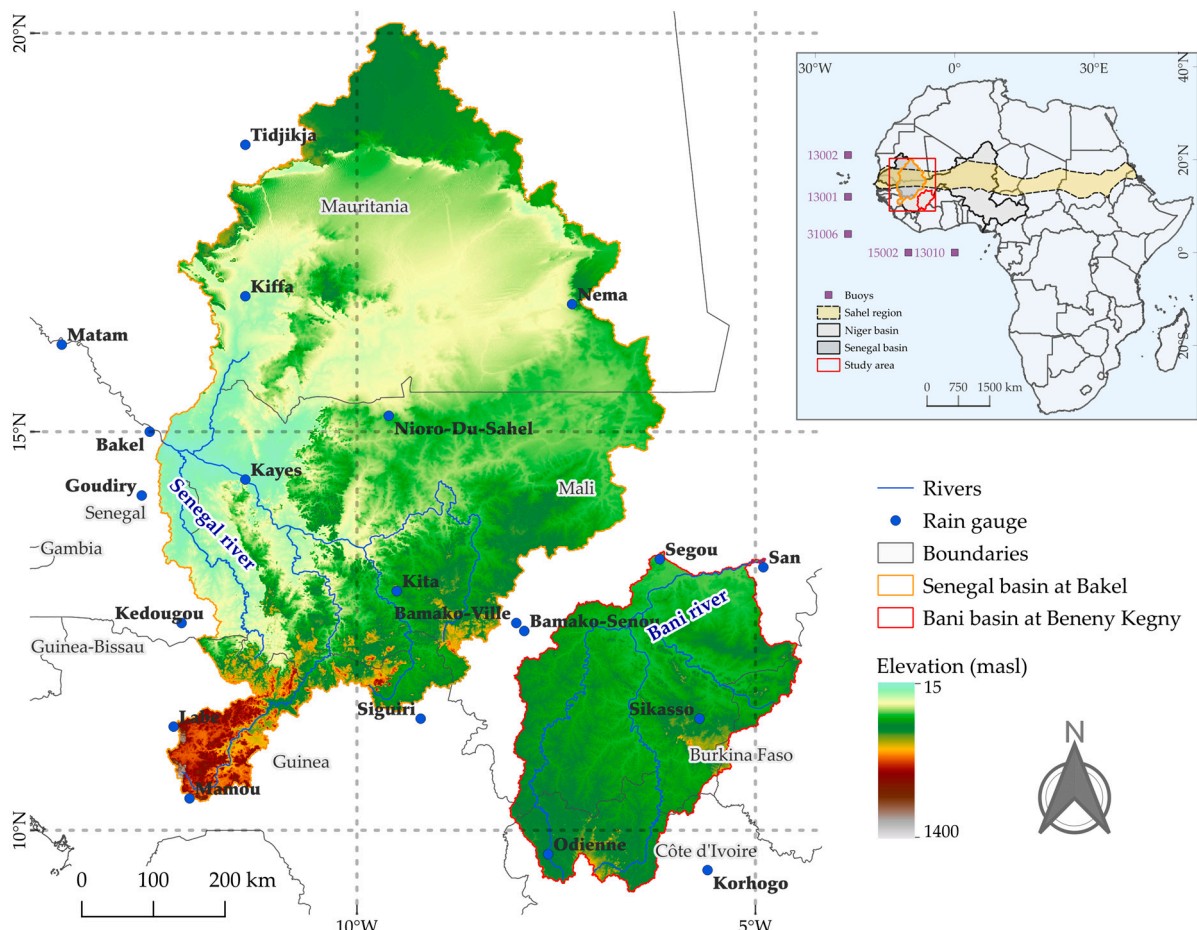

**Figure 1.** Location of the basins of the Bani River at Beneny Kegny (upper Niger) and the Senegal River at Bakel on a digital elevation model (DEM). Location of rain gauges (blue dots) and buoys in the Atlantic located off the coast of West Africa (magenta boxes) and the Sahel region (yellow stripe).

Delimitation of the basins was performed based on a digital elevation model (DEM) of 3 arcs of a second (0.000833°, ~90 m) of spatial resolution, taken from the Shuttle Radar Topography Mission (SRTM) [19], available at https://srtm.csi.cgiar.org/, accessed on 20 December 2018. In a study conducted by Bâ et al. [20] on the Senegal River, a ~1 km resolution DEM was used to delimit the Senegal basin. This study revealed that the drainage areas up to the Bakel and Kayes hydrometric gauges are larger than those reported by the official map of the Organisation pour la Mise en Valeur du fleuve Sénégal (OMVS). However, the area that was not considered by the OMVS is in the northern part of the watershed located in the Sahara Desert, which does not contribute to the runoff.

The Bani River is the main tributary of the Niger River. The Bani basin at the Beneny Kegny hydrometric gauge has an approximate area of 112,000 km² [8]. The basin's topography (Figure 1) is characterized by an elevation between 270 and 760 masl. The slope varies between 0% and 10%, and the average slope of the basin is 0.85%. On average (1981–2000),

the annual rainfall varies between 1250 mm in Odienne (south of the basin at 400 masl) to 615 mm in Segou (north of the basin at 280 masl) [21]. According to the GlobCover [22], the land cover is mainly characterized by crop mosaics (Mosaic Cropland), lush vegetation (grassland, shrubland, forest), and small shrubs, while in a small part of the south, the ground is covered by forest [23].

The Senegal River basin at Bakel has an approximate area of 440,000 km$^2$, its altitude range varies between 15 and 1400 masl, the terrain's slope varies between 0% and 19% and the average slope is 1.08% (very gently sloping). Annual rainfall is about 80 mm year$^{-1}$ in the northern part of the basin, while at the southern edge of the basin it can reach 2000 mm year$^{-1}$ [8]. According to the GlobCover [22], the land cover in the north of the basin is characterized by a lack of vegetation (bare areas), the center of the basin is covered by brushwood, crop mosaics (Mosaic Cropland) and lush vegetation (grassland, shrubland and forest), while the southern part of the basin is covered by evergreen forests (broadleaved evergreen forest) and semi-deciduous forest [23].

*2.2. Input Data*

In several countries of the region, meteorological data are not free and the available data records lack continuity and reliability [8,24]. In some weather stations, daily rainfall has been recorded since the beginning of the last century, but data from recent years are not accessible. The most recent data to which we had access have observations prior to 2015. Analysis shows these data have a suitable quality [8,25].

Recent studies have shown that rainfall data from satellite products provide suitable results in this region [8,25,26]. For example, the PERSIANN-CDR product [17] has records from 1983 to present at 0.25° of spatial resolution and daily frequency. Monthly PERSIANN-CDR data have been evaluated by Bâ et al. [8] in 18 rain gauges located in West Africa and then used the same daily rainfall for flow simulation. The results of the evaluation were satisfactory ($R^2 > 0.8$) at most gauges. PERSIANN-CDR data can be found at the website of the Center for Hydrometeorology and Remote Sensing (CHRS) at the University of California, Irvine (https://chrsdata.eng.uci.edu/, accessed on 16 September 2018).

For this study, rainfall of 20 rain gauges were used (Figure 1) to evaluate the PERSIANN-CDR data. A total of 456 monthly images of PERSIANN-CDR product between 1983 and 2020 were used and evaluated in the concomitant period. The rainfall data are obtained from the Meteorological Services of Mauritania, Senegal, Guinea, Mali, and Cote d'Ivoire. These data are used only in the validation process of satellite estimated products.

The historical PERSIANN-CDR data are used as the dependent variable in the forecasting models. The basins of the Bani and Senegal Rivers are made up of 725 pixels of 0.25° × 0.25°. In addition, data from the Atlantic Ocean, sea surface temperature (SST), mean sea level pressure (MSLP), relative humidity (RHUM), and 5 Pacific indices Niño1 + 2, Niño3.4, Niño4, ONI, and TNI are used as explanatory variables of the models.

SST is the main variable in the rainfall forecast [2,13]. In the forecast models, the Tropical Atlantic (70°W, 20°E, 20°S, 40°N) was used as the predictor variable of the linear, polynomial, and exponential models. In addition, the MSLP and RHUM data (in the same extension) and the El Niño Southern Oscillation (ENSO) indices were included in a multivariate model. These indices are based on the SST anomalies of tropical Pacific regions, Niño1 + 2 (90°W to 80°W, 0° to 10°S), Niño3.4 (170°W to 120°W, 5°N to 5°S), Niño4 (160°E, 150°W, 5°S, 5°N), ONI (170°W to 120°W, 5°N to 5°S), and TNI (Niño1 + 2 and Niño3.4) [27].

The reanalysis data of the SST product of ERA5 were used at 0.25° × 0.25° of spatial resolution and daily frequency [28]. These data come from two providers. Before September 2007, SST data are from the Group for High-Resolution Sea Surface Temperature (GHRSST) of the National Centers for Environmental Information (NCEI). As of September 2007, data from the Operational Sea Surface Temperature and Sea Ice Analysis (OSTIA) [29] were used. Data from ERA5: SST, MSLP, and RHUM are available at https://cds.climate.copernicus.eu [28], and the NCAR/UCAR Climate Date Guide El Niño indices are available at https://

climatedataguide.ucar.edu/climate-data/nino-sst-indices-nino-12-3-34-4-oni-and-tni [27], accessed on 25 September 2018.

Sea temperature data from 5 buoys are used to validate the SST of the Atlantic Ocean. These data were obtained from The Prediction and Research Moored Array in the Tropical Atlantic (PIRATA) of the National Data Buoy Center, The National Oceanic and Atmospheric Administration (NOAA) [30]. Buoy data are available at https://www.ndbc.noaa.gov/, accessed on 26 September 2018. Buoys located near the coast of West Africa that have continuous records were chosen (Figure 1). The assessment, on one hand, of the PERSIANN-CDR and SST products and, on the other hand, of the forecast models, is carried out at point-to-pixel. That is, the rainfall of the rain gauges and the temperature of the buoys are compared with the data of the raster cells where the gauges/buoys are located.

*2.3. Ocean-WAM Teleconnections*

It is known that the Atlantic Ocean is the main source of humidity for West Africa [2,13], but it is necessary to know which region is better tele-connected between ocean-atmospheric variables and WAM. In addition, it is important to find the optimal time lag between the variables and the WAM. In previous studies, authors divide SST using some techniques such as Folland et al. [4] in a study on the Sahel, they grouped pixels of the SST anomalies into 10° x 10° cells. Gado et al. [31] and Sittichok et al. [13] decreased the number of SST components of the Atlantic using PCA and canonical correlation analysis (CCA).

Phenomena with a social impact on the region, such as droughts and floods, are related to the tropical Atlantic variability (TAV). The TAV has interactions, particularly with trade wind fluctuations, SST, and precipitation [32]. The SST is tele-connected with rainfall that varies between 0, 6, and 12 months depending on the place in the ocean [13,31]. In this study, PCA and cluster analysis were used, with the purpose of grouping pixels of the Atlantic SST homogeneous values and finding the region with the best teleconnection between ocean variables and rainfall of the Bani and Senegal River basins. The PCA performs an orthogonal test transformation on the 456 monthly images between 1983 and 2020. The result is a reduced set of synthetic explanatory covariates, called principal components, which are not correlated to each other [33,34]. Clustering was applied to group the pixels with maximum homogeneity in each group. The k-means method was used to find the differences between groups [34]. The cross-correlation analysis helped to find the optimal lag between the predictors (SST, RHUM, MSLP, El Niño indices) and the PERSIANN-CDR precipitation.

*2.4. Forecasting Models*

Linear and non-linear models are commonly used to find the relationship of predictors or explanatory variables to the response variable [2]. The linear, second-order polynomial, stepwise, and exponential models are defined in Equation (1) through Equation (4), respectively.

$$\hat{Y} = b_0 + b_1\left(SST_{(lag_{SST})}\right) \tag{1}$$

$$\hat{Y} = b_0 + b_1\left(SST_{(lag_{SST})}\right) + b_2\left(SST_{(lag_{SST})}\right)^2 \tag{2}$$

$$\hat{Y} = b_0 + b_1 SST_{lag_{SST}} + b_2 MSLP_{lag_{MSLP}} + b_3 RHUM_{lag_{RHUM}} + b_4 ElNi\tilde{n}o1 + 2_{lag_{El\ni\tilde{n}o1+2}}$$
$$+ b_5 ElNi\tilde{n}o3.4_{lag_{\tilde{n}o3.4}} + b_6 ElNi\tilde{n}o4_{lag_{ElNi\tilde{n}o4}} + b_7 ONI_{lag_{ONI}} + b_8 TNI_{lag_{TNI}} \tag{3}$$

$$\hat{Y} = b_0 + e^{\left(SST_{lag_{SST}}{}^{b_1}\right)} \tag{4}$$

where $\hat{Y}$ is the forecast precipitation, $b_0, b_1, \ldots, b_n$ are the coefficients of the models and $e$ is Euler's number, $SST_{lagSST}$, ..., $TNI_{lagTNI}$ are the ocean-atmospheric covariables with their respective lag. The coefficients are obtained by the method of least squares; in the case of the exponential model, they are obtained by trial and error in an iterative process.

Eight covariates were used in the stepwise regression model. The collinearity analysis, measured with the variance inflation factor (VIF) suggests reducing the number of variables to decrease variance inflation and avoid overfitting the model [35].

Rainfall forecasts in the 725 pixels of the study area are calculated using the SST data of the Atlantic regions. Models of Equations (1)–(4) are applied in each of the pixels that make up the basins of the Bani and Senegal Rivers. The results of Equations (1), (2) and (4) of each SST region of the Atlantic are compared with the PERSIANN-CDR value of each pixel (dependent variable). For the stepwise regression model, Equation (3), a combination was made between all the predictors considering the lag of each of them.

Covariables series and the dependent variable (PERSIANN-CDR) were divided into two samples, 70% for models' development and 30% for model validation. This procedure was carried out using an R script (see supplementary materials: https://github.com/lebalcazar/sahel). About 600 iterations have been performed. The parameters of the non-linear model, Equation (4), are calibrated by trial error, once these converge, the best model parameters are obtained. In the R script, a maximum of 1000 iterations are restricted to obtain the parameters of the exponential model; however, it was observed that they converge after 100 iterations.

### 2.5. Model Assessment

Once the models are obtained for each pixel, an assessment is performed with the validation sample for each pixel. Next, forecasts rainfall are compared with the observed data using objective criteria such as adjusted coefficient of determination ($R^2$adj), Equation (5) and the Akaike information criterion (AIC), Equation (6). Models with significant parameters (*p*-value < 0.05) are chosen, $R^2$adj > 0.5 and lower AIC value are chosen, as well as models that have a lag greater than or equal to six months, reasonable time for authorities to take preventive measures [31].

$$R^2 adj = 1 - \frac{n-1}{n-k-1}\left(1 - R^2\right) \tag{5}$$

where *n* is the number of observations in the sample, k is the number of model variables and $R^2$ is the coefficient of determination, Equation (7). $R^2$adj indicates the degree of effectiveness of independent variables in explaining the response variable.

Increasing of independent variables escalate the value of the quotient. $(n-1)/(n-k-1)$. $R^2$ is reduced as a function of the increment of variables; therefore, $R^2$adj penalizes the addition of coefficients in the model [36].

$$AIC = 2k - 2ln(L) \tag{6}$$

where k is the number of variables in the model and L is the maximum likelihood value for the estimated model.

The AIC proposed by Akaike [37] is used in model selection. This criterion considers the goodness of fit and the complexity of the model, based on the penalty for the number of explanatory variables used. For example, a model with a larger number of explanatory covariates increases the probability of having a better fit, however, this can result in an overfitting of the model and is penalized by the AIC.

On the other hand, the coefficient of determination ($R^2$) and Nash–Sutcliffe efficiency coefficient (NSE) were obtained [38]. In addition, error was calculated with percent bias (PBIAS, [39]), relative error (RE), and mean absolute error (MAE) [40].

$$R^2 = \left( \frac{\sum_{i=1}^{n}\left(obs_1 - \overline{obs}\right)\left(sim_i - \overline{sim}\right)}{\left(\sum_{i=1}^{n}\left(obs_i - \overline{obs}\right)^2\right)^{1/2}\left(\sum_{i=1}^{n}\left(sim_i - \overline{sim}\right)^2\right)^{1/2}} \right)^2 \tag{7}$$

$$NSE = 1 - \frac{\sum_{i=1}^{n}(obs_i - \text{sim}_i)^2}{\sum_{i=1}^{n}\left(obs_i - o\bar{b}s\right)^2} \tag{8}$$

$$PBIAS = \frac{\sum_{i=1}^{n}(\text{sim}_i - obs_i)}{\sum_{i=1}^{n}obs_i} \times 100 \tag{9}$$

$$RE = \left|\frac{obs_i - \text{sim}_i}{obs_i}\right| \times 100 \tag{10}$$

$$MAE = \frac{\sum_{i=1}^{n}|obs_i - \text{sim}_i|}{n} \tag{11}$$

where $obs_i$ and $\text{sim}_i$ are, respectively, the observed and simulated variable of the month i, $o\bar{b}s$ and $\bar{\text{sim}}$ are, respectively, observed and simulated mean, and *n* is the amount of data.

$R^2$ measures the proportion of the variance explained by the model. $R^2$ range varies between 0 and 1, with 1 being the optimal value, and values greater than 0.50 are considered acceptable [41].

The NSE is used to determine the relative magnitude of the residual variance and the variance of the observations. NSE coefficient varies between $-\infty$ and 1, when NSE is equal to 1, it indicates a perfect simulation of the model. If the NSE equals 0, it indicates that the predictions of the model are as accurate as the mean of the observed data. A negative NSE indicates that the observed mean is a better predictor than the model [40,41].

PBIAS is used to determine how well the model simulates the average magnitudes for the output response of interest. PBIAS is useful for long-term continuous simulations and allows to identify the average bias of the model simulations. PBIAS range varies between $-\infty$ and $\infty$, 0 is the optimal value. Positive values indicate that the model overestimates the observed rainfall and negative values indicate that the model underestimates the rainfall [39].

RE is the quotient between the absolute error of the simulated rainfall and the observed rainfall. This allows to understand the performance of the model among different responses. In addition, the differences between the observed and simulated values are quantified as relative deviations. This significantly reduces the influence of absolute differences during peaks [40].

MAE measures the error of the values calculated by the model. It is calculated and presented in the same unit of the model; therefore, it is easier to interpret. In addition, it is very useful in long-term continuous simulations [40]. MAE usually has a magnitude that is equal to or less than RMSE; however, it gives greater weight to the peaks, so adjustments must be made using the standard deviation of the observations [41].

## 3. Results

The most relevant results obtained in this study are presented below.

### 3.1. Validation of Satellite Products

The process begins with the comparison between the monthly PERSIANN-CDR data and the monthly gauged rainfall, followed by the comparison between ERA5 monthly mean temperature and observed monthly mean temperature at 5 Atlantic buoys, Tables 1 and 2, respectively.

Results of the assessment of PERSIANN-CDR and SST products were satisfactory considering the statistical criteria $R^2$, PBIAS, and MAE. PERSIANN-CDR yields in 20 rain gauges (1983–2014) were between satisfactory to very good ($0.510 \leq R^2 \leq 0.879$; $0.1 \leq |PBIAS| \leq 25.0$; $3.0 \leq MAE \leq 58.0$) (Table 1). PERSIANN-CDR assessments are consistent with those obtained by Bâ et al. [8] in 18 rain gauges (1995–2015). The SST assessment in 5 Atlantic buoys was also very good when comparing the SST ERA5 product with the observed values in the Atlantic buoys between 1997 and 2019 ($0.928 \leq R^2 \leq 0.991$; $0.60 \leq |PBIAS| \leq 1.40$; $0.20 \leq MAE \leq 0.38$) (Table 2).

**Table 1.** Validation of monthly precipitation of the PERSIANN-CDR product with the observed data of 20 rain gauges between 1983 and 2014.

| Rain Gauge | $R^2$ | PBIAS (%) | MAE (mm) | N° of Data |
|---|---|---|---|---|
| Tidjikja | 0.672 | 13.2 | 10.0 | 104 |
| Kiffa | 0.600 | −0.1 | 21.0 | 115 |
| Nema | 0.510 | 9.0 | 23.0 | 116 |
| Matam | 0.538 | 2.3 | 28.0 | 120 |
| Nioro-Du-Sahel | 0.764 | 15.4 | 24.0 | 122 |
| Bakel | 0.756 | 3.2 | 30.0 | 159 |
| Kayes | 0.779 | 16.7 | 29.0 | 122 |
| Goudiry | 0.688 | 18.2 | 36.0 | 140 |
| Segou | 0.833 | 11.0 | 26.0 | 169 |
| San | 0.858 | 2.8 | 22.0 | 168 |
| Kita | 0.879 | 13.7 | 30.0 | 142 |
| Kedougou | 0.682 | 4.7 | 51.0 | 156 |
| Bamako-Ville | 0.817 | 0.8 | 34.0 | 140 |
| Bamako-Senou | 0.838 | 8.2 | 30.0 | 183 |
| Labe | 0.825 | 25.0 | 58.0 | 204 |
| Siguiri | 0.784 | 9.8 | 40.0 | 190 |
| Sikasso | 0.85 | −1.7 | 29.0 | 202 |
| Mamou | 0.815 | 11.1 | 49.0 | 246 |
| Odienne | 0.827 | 4.1 | 35.0 | 120 |
| Korhogo | 0.717 | −9.2 | 38.0 | 95 |

**Table 2.** Validation of the SST ERA5 with monthly mean temperature in 5 Atlantic buoys between 1997 and 2019.

| Buoys | $R^2$ | PBIAS (%) | MAE (°C) | N° of Data |
|---|---|---|---|---|
| 13001 | 0.977 | −1.30 | 0.36 | 127 |
| 13002 | 0.991 | −0.80 | 0.23 | 128 |
| 13010 | 0.983 | −1.40 | 0.38 | 194 |
| 15002 | 0.987 | −0.90 | 0.27 | 198 |
| 31006 | 0.928 | −0.60 | 0.20 | 137 |

### 3.2. Classification of the Atlantic Variables

SST, RHUM, and MSLP Atlantic data were processed (456 monthly raster images of each variable between 1983 and 2020). The PCA and cluster analysis allowed us to divide the Tropical Atlantic into homogeneous regions. The PCA reduced the set of 456 images into two principal components that explain 97.3% of the SST variance, 96.0% of the MSLP variance, and 98.0% of the RHUM variance. The principal components were used in the cluster analysis. The k-means method and the silhouette method suggest that the optimal number of clusters is k = 3 for SST and RHUM and k = 2 for MSLP (Figure 2).

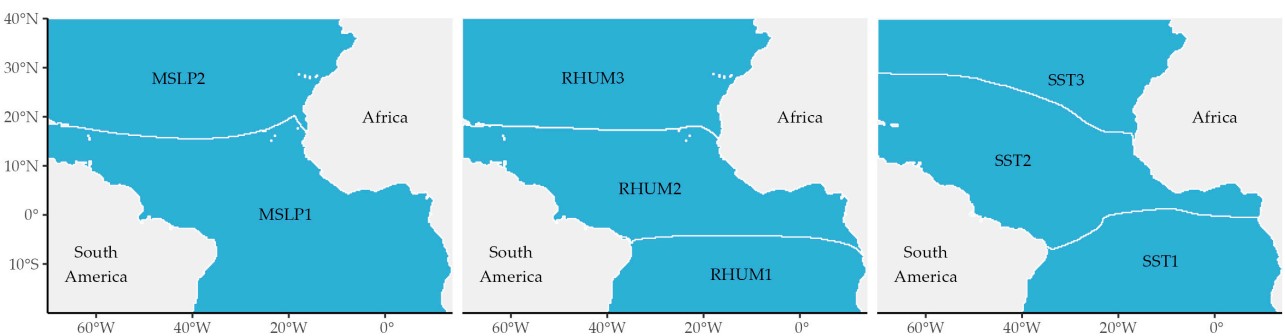

**Figure 2.** Classification of the Tropical Atlantic into homogeneous regions, using PCA and cluster analysis: SST1, SST2, SST3, RHUM1, RHUM2, RHUM3, MSLP1, and MSLP2.

### 3.3. Selection of Forecast Models

SST of the 3 Atlantic regions (Figure 2) was used in the linear, polynomial, and exponential models. Cross-correlation analyses showed that SST has a higher correlation with time lags of 5, 10, and 11 months, respectively, for the SST1, SST2, and SST3 regions. Then, in the stepwise regression model, the covariables of the SST, RHUM, and MSLP regions were used (Figure 2), and the indices of the Pacific Niño1 + 2, Niño3.4, Niño4, ONI, and TNI.

It was observed that the SST is the most influential variable in the WAM and the coefficient of determination between PERSIANN-CDR and SST in a linear model is greater than 0.70. When the models are applied for each pixel, it is observed that the polynomial model reproduces better rainfall in the Bani and Senegal River basins, followed by the stepwise regression model. The highest performance was obtained with the polynomial model and SST3, the north Tropical Atlantic region, and with a lag of 11 months (NSE = ~0.80). In the south of the basins, at Mamou, Labe, Odienne, and Korhogo rain gauges, yields were better with the stepwise regression model with the SST3 region; however, the AIC difference between these models is negligible. On the other hand, in the north of the Senegal basin, no model was able to reproduce rainfall with the selection criteria ($p$-value $< 0.05$, $R^2$adj $> 0.5$, and lower AIC).

To compare the results of the stepwise regression model in the three regions, the covariates that contribute to improving the explanation of the variance of the phenomenon are added to the SST. This model provided better results in the SST3 region with an $R^2$adj of 0.845; that is, more than 84% of the precipitation variability is explained by the multivariate model. However, to avoid overfitting the model, the VIF analysis [35] suggests using only the SST, RHUM, Niño1 + 2, and TNI covariates.

In the northwest of the Senegal River basin at Bakel, Matam, and Kiffa rain gauges, some pixels were better with the polynomial model and the SST1 region (Gulf of Guinea); however, the lag of these simulations is 5 months, which is less than the objectives set in this research. When comparing the AIC and $R^2$adj between the polynomial model with SST1 and SST3 regions, the difference is negligible (Figure 3).

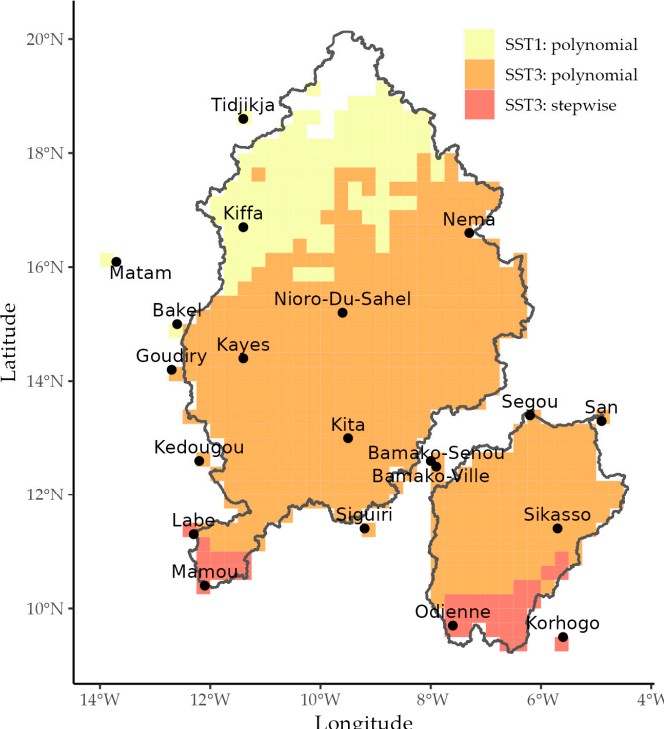

**Figure 3.** Spatial distribution of models that best reproduce rainfall in each pixel of the Bani and Senegal River basins.

Table 3 shows that the polynomial model yields the highest NSE coefficient values, followed by the stepwise regression model, the exponential model, and finally, the simple linear model. In all cases, no model was able to predict rainfall in the northern Senegal River basin. To have the rainfall forecast over the entire area of the basins, restrictions of the selection criteria were removed: $p$-value < 0.05, $R^2$adj > 0.5, AIC less, and lag greater than or equal to six months. Figure 4 shows the performance of the polynomial model, measured with the NSE, in the SST1, SST2, and SST3 regions. NSE is found to be higher with SST of the North Tropical Atlantic (SST3) and an 11-month lag, especially in the south of the basins, which is the most humid part of the studied basins.

**Table 3.** Performance of precipitation forecast models, measured with NSE: linear, exponential, polynomial, and stepwise regression models at pixel-to-point.

| Rain Gauges | Models | | | |
|---|---|---|---|---|
| | Linear (lm) | Polynomial (Poly) | Exponential (nls) | Stepwise Regression |
| Tidjikja | - | - | - | - |
| Kiffa | 0.509 | 0.629 | 0.627 | 0.610 |
| Nema | 0.541 | 0.685 | 0.674 | 0.585 |
| Matam | 0.501 | 0.673 | 0.674 | 0.617 |
| Nioro-Du-Sahel | 0.647 | 0.767 | 0.707 | 0.714 |
| Bakel | 0.645 | 0.752 | 0.677 | 0.676 |
| Kayes | 0.682 | 0.754 | 0.705 | 0.741 |
| Goudiry | 0.730 | 0.820 | 0.726 | 0.792 |
| Segou | 0.696 | 0.843 | 0.839 | 0.703 |
| San | 0.743 | 0.879 | 0.873 | 0.765 |
| Kita | 0.782 | 0.872 | 0.838 | 0.802 |
| Kedougou | 0.762 | 0.821 | 0.790 | 0.821 |
| Bamako-Ville | 0.772 | 0.865 | 0.855 | 0.791 |
| Bamako-Senou | 0.770 | 0.833 | 0.820 | 0.794 |
| Labe | 0.836 | 0.853 | 0.812 | 0.837 |
| Siguiri | 0.816 | 0.860 | 0.826 | 0.837 |
| Sikasso | 0.869 | 0.905 | 0.888 | 0.889 |
| Mamou | 0.815 | 0.815 | 0.755 | 0.842 |
| Odienne | 0.786 | 0.808 | 0.787 | 0.817 |
| Korhogo | 0.776 | 0.776 | 0.731 | 0.828 |

- Not estimated by models.

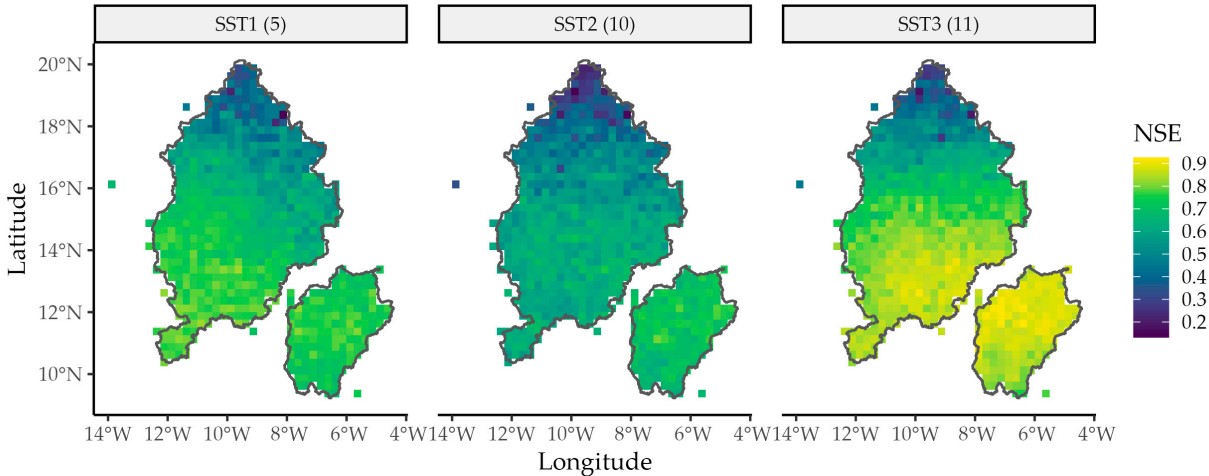

**Figure 4.** Spatial distribution of NSE with the polynomial model in the 3 SST regions of the Atlantic and their respective time lags (5 months for SST1, 10 months for SST2, and 11 months for SST3).

Rainfall forecasts with the polynomial model (Figure 4) adequately reproduce seasonal rainfall of the WAM. Figure 5 shows the spatio-temporal distribution of the rainfall forecast. It shows that it is temporarily distributed between May and October, mainly in the south of the Bani and Senegal basins. In July and August, the rainfall is distributed throughout the basins' area, with a decrease in rainfall as latitude increases. Rainfall forecasts from 1984 to 2020 are presented in Table A1, Appendix A (as complementary material). The average rainfall in the period 1984–2020 is ~690 mm year$^{-1}$, the minimum is ~490 mm in 1985, and the maximum is ~830 mm in 2016.

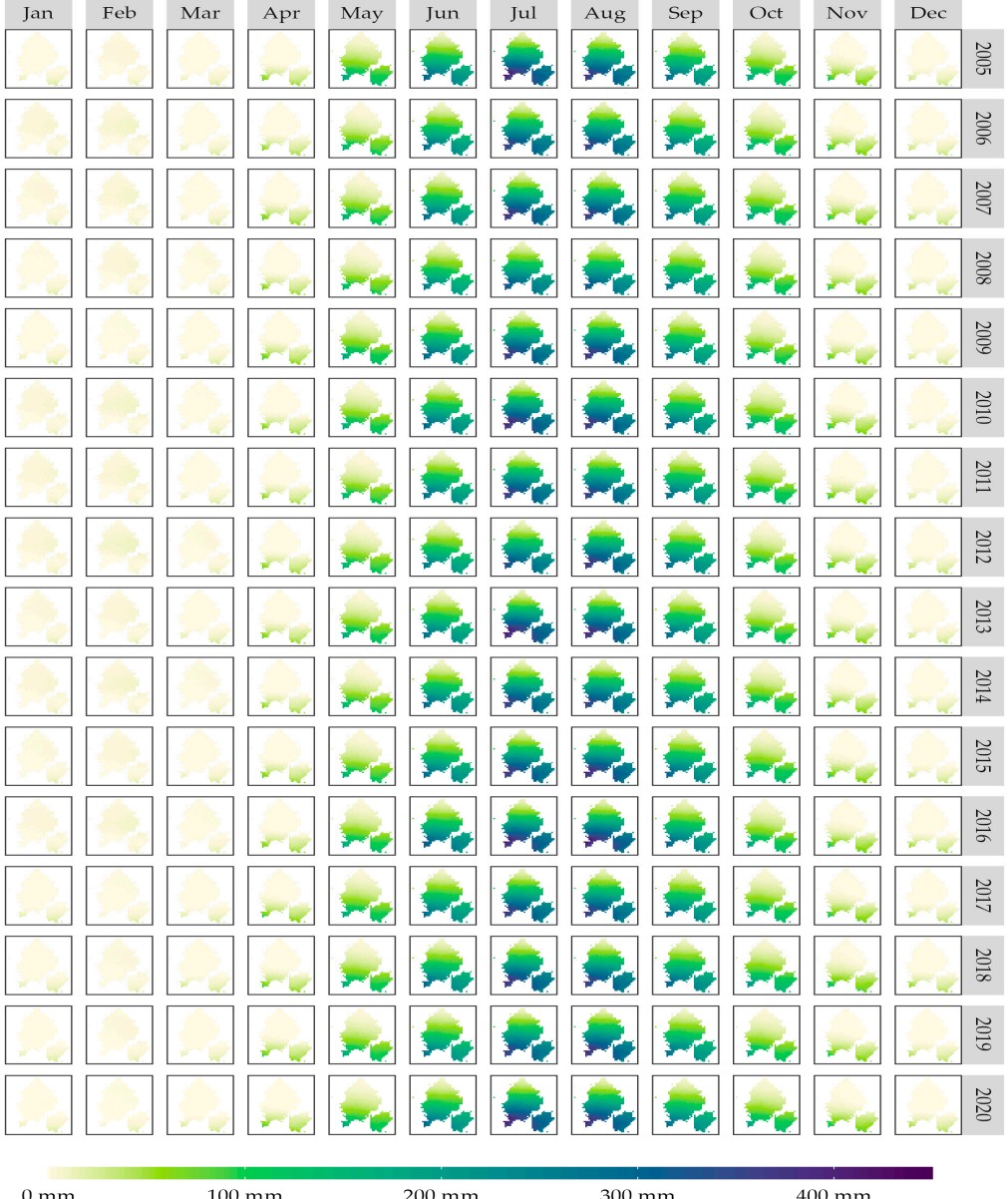

**Figure 5.** Spatial-seasonal distribution of rainfall forecast with the polynomial model, sample from the years 2005 to 2020.

### 3.4. Comparison of the Forecasts with the Reference Rainfall

According to the PRESASS forecast [14], by 2021, summer rainfall in the Sahel and Sudan regions was expected to be equivalent to or higher than normal. In addition, rainfall was predicted to have an early to normal start and a late to normal end. As historical PERSIANN-CDR data are available, these records were used to obtain the normal precipitation (1991–2020) as in PRESASS [15].

The results of this study were compared with those of PRESASS for 2021 as an example. The gamma distribution function was fitted to each PERSIANN-CDR monthly precipitation sample of each pixel to determine the quantiles at three characteristic values of probability of non-exceedance. Figure 6 presents the monthly frequency hyetograph (MFH) at pixels where a rain gauge is located for the rainy season (May to October) and rainfall of the year 2021. Rainfall quantiles were computed for probabilities of 0.30, 0.50, and 0.70, representing, respectively, under normal, near normal, and above normal.

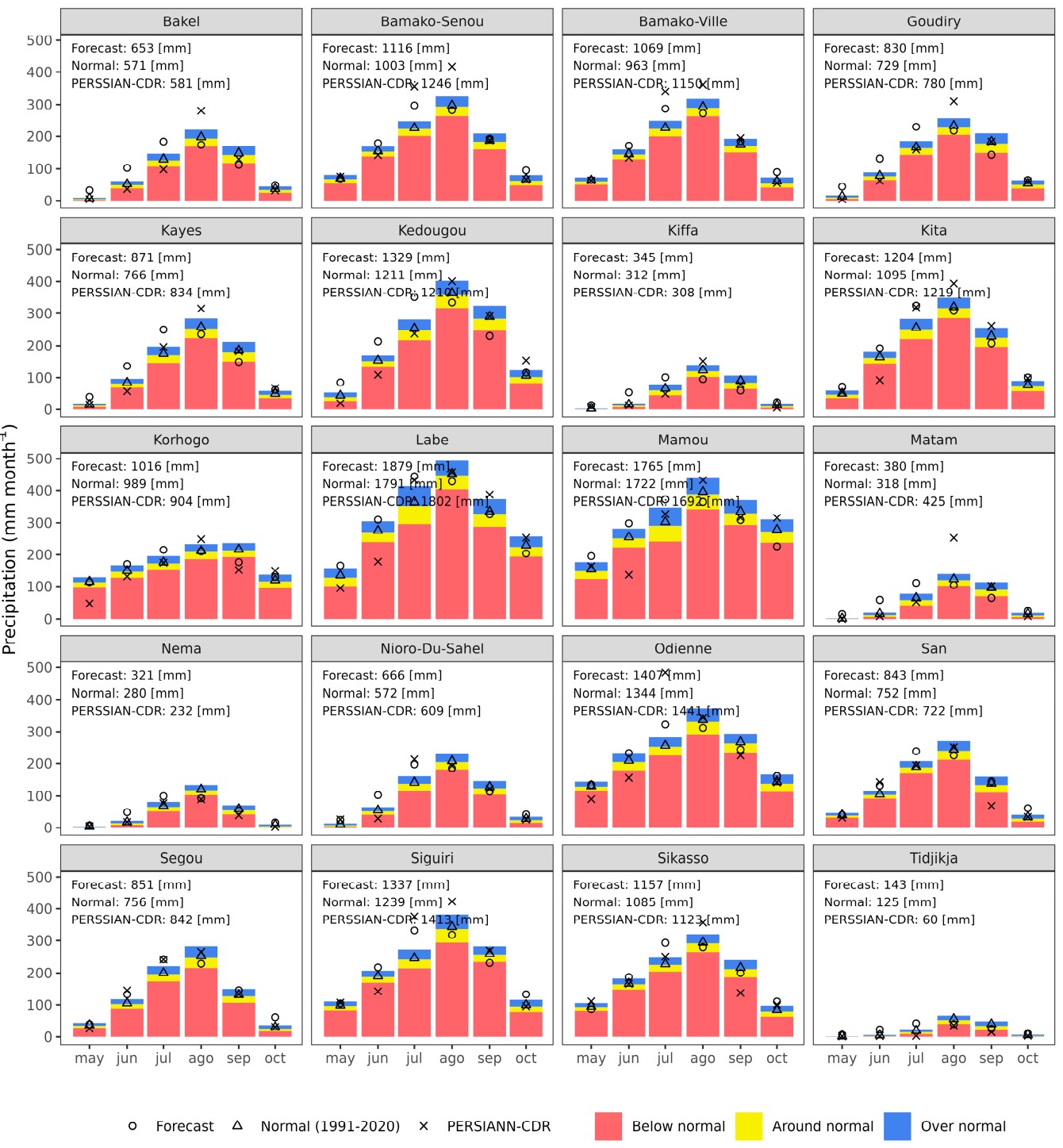

**Figure 6.** Comparison between the forecast for 2021 and PERSIANN-CDR monthly rainfall (mm) using the monthly frequency hyetograph (MFH) of the pixel where the rain gauge is located.

At the beginning of the rainy season, and for all regions, forecasts are around normal and almost equivalent to observed rainfall (PERSIANN-CDR). From May to October, forecasts are around normal or over normal in most pixels. In general, the 2021 forecast is consistent with the forecast of PRESASS [14].

When comparing the average rainfall of the forecast for the year 2021 with the normal precipitation, 90% of the area corresponds to wet regions (Figure 7). In the south region of the Bani and Senegal River basins and in the northern end of the Senegal River basin, the forecast is classified as around normal.

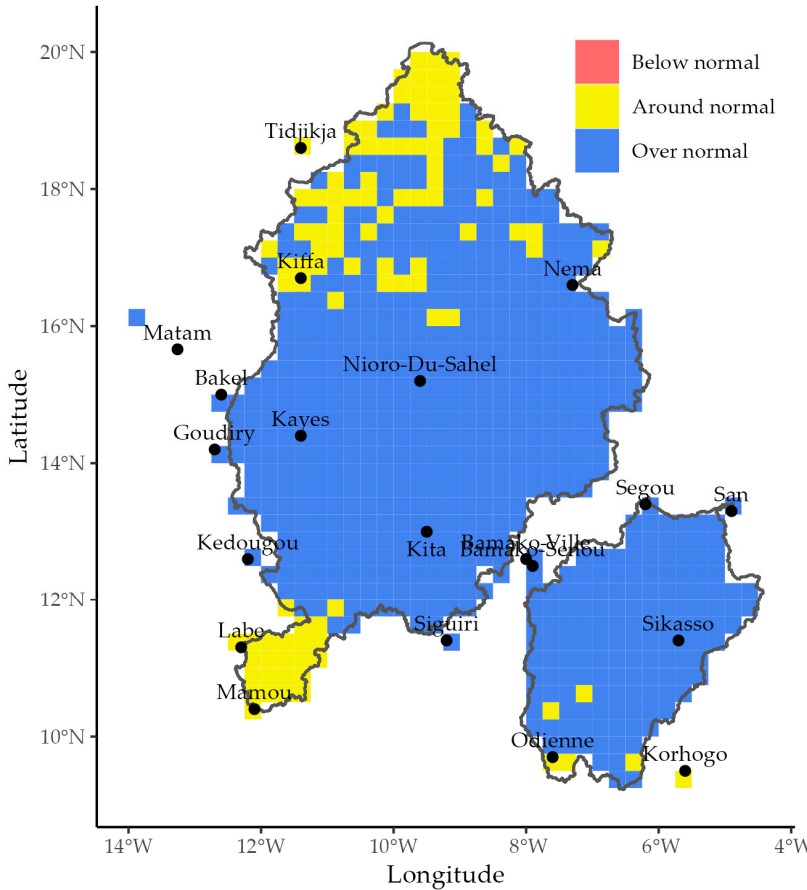

**Figure 7.** Classification of the seasonal rainfall forecast for 2021 over the Bani River and Senegal River watersheds in terms of wet, normal, and dry.

Table 4 shows the comparison between the rainfall forecasts (May–October) with the PERSIANN-CDR for the years 2017–2021. It was found that the forecasts (FRC) are lower but close to the observed rainfall (PERSIANN-CDR). The relative error varies between 1.1% and 38%. However, in the northern end of the Senegal River basin (such as Tidjikja, see Table A2, Appendix A), the relative error can exceed 100% because the models do not simulate rainfall well enough in this area.

**Table 4.** Validation of the forecasted rainfall (FRC) in a 4-pixel sample with the "observed" rainfall PERSIANN-CDR (CDR) at the point-to-pixel from the last 5 years.

| Year | Labe | | | Nema | | | Nioro-Du-Sahel | | | Segou | | |
|---|---|---|---|---|---|---|---|---|---|---|---|---|
| | CDR | FRC | RE | CDR | FRC | RE | CDR | FRC | RE | CDR | FRC | RE |
| 2017 | 1726.7 | 1815.4 | 5.1 | 312.4 | 295.5 | 5.4 | 571.8 | 618.2 | 8.1 | 781.7 | 795.1 | 1.7 |
| 2018 | 1595.0 | 1863.4 | 16.8 | 380.8 | 311.4 | 18.2 | 689.4 | 648.0 | 6.0 | 759.5 | 830.5 | 9.3 |
| 2019 | 1989.0 | 1767.4 | 11.1 | 262.7 | 282.8 | 7.7 | 661.4 | 593.7 | 10.2 | 907.8 | 765.4 | 15.7 |
| 2020 | 1682.7 | 1859.6 | 10.5 | 336.5 | 315 | 6.4 | 772.9 | 653.5 | 15.4 | 910.3 | 836.2 | 8.1 |
| 2021 | 1802.5 | 1879.0 | 4.2 | 231.9 | 321.4 | 38.6 | 609.3 | 665.5 | 9.2 | 841.8 | 850.6 | 1.0 |

We measure the performance of the forecast model on each pixel, a sample of 4 pixels where the weather stations are located. The errors for the remaining pixels are presented in Table A2, Appendix A.

## 4. Discussion

### 4.1. Rainfall Distribution

In West Africa, rainfall has a negative gradient related to latitude. In the south, in the upper part of the Bani and Senegal River basins, precipitation is about 2800 mm year$^{-1}$, while in the north, it is about 100 mm year$^{-1}$ [8]. The humidity of West Africa is almost entirely caused by WAM. The wet period is from May to October in the south of the basins [4,6], and the rainiest months are July–August–September, in the area [7,8].

Previous studies in the Sahel report several hypotheses to describe the drought that occurred in the last decades [31,42,43]. The most significant ones are listed below. Cooling of the SST in the north Tropical Atlantic and sudden warming in the south leads to the migration of the ITCZ further south, causing lower humidity in the region [2,9,32]. In addition, the associated events such as El Niño and La Niña influence rainfall patterns of tropical areas [42].

Satellite products were validated with observations in situ. PERSIANN-CDR [17] precipitation and temperature product data were compared with the rain gauge data at pixel-to-point, and the coefficient of determination was calculated. Results are consistent with those reported in previous studies in the same region [8,24,25]. It is important to highlight that the SST data were validated with observations on the Atlantic buoys at pixel-to-point. The results were very good, and the R$^2$ was higher than 0.93.

### 4.2. Atlantic Regions

Statistical methods of PCA and cluster analysis were used to regionalize SST, RHUM, and MSLP of the Tropical Atlantic. Grouping into homogeneous regions optimizes the computing time of the forecasting processes. In recent studies, different techniques have been used to reduce the spatial-temporal dimension of the variables. For instance, Sittichok et al. [13] used PCA methods, canonical correlation, and stepwise regressions between the Atlantic and Pacific to reduce the number of components and select the optimal lag. Folland et al. [4] grouped the 2.5° × 2.5° SST into 10° × 10° cells to reduce the number of components. Studies carried out by Gado Djibo et al. [2,31] used the method proposed by Sittichok et al. [12,13] to estimate optimal lag. In addition, they used Bayesian algorithms to detect change points for the purpose of combining models with dynamic parameters.

### 4.3. Rainfall Forecasts

The performance of the linear and exponential models evaluated with the NSE criteria provided values in the ranges of (0.509, 0869) and (0.627, 0.888), respectively, for the 20 pixels where rain gauges are located (Table 3). For these pixels, the NSE results of the stepwise regression model were between 0.585 and 0.889. Sittichok et al. [13] also used a stepwise model and obtained an NSE of 0.387 (Table 5).

Rainfall estimates with the polynomial model and the SST of the tropical north Atlantic yielded the best results. NSE varies between 0.629 and 0.905, with a lag of 11 months. Forecasts were better in the southern basins of the Bani and Senegal Rivers. However, in the northern part of the Senegal River basin (desert area), none of the four models was able to reproduce the rainfall with the selection criteria. To have one model for each pixel of this dry region, restrictions on the criteria were removed, and the models with the lowest AIC were selected. It seems that SST alone cannot explain the dynamics of precipitation in the Sahara. This region is characterized by very low annual rainfall (<100 mm). Several factors may cause this permanent drought. The high temperatures mean that the sea humidity does not reach the area. The extremely overheated winds constitute barriers to this arrival of humidity. Other factors should be considered for the seasonal rainfall forecasting in this area.

**Table 5.** Comparison of results of the precipitation forecast models of this study with previous studies.

| Author | Data/Location | Sources and Characteristics | Models (Rainfall Forecast) | Lag (Months) | Efficiency |
|---|---|---|---|---|---|
| In this study | Atlantic SST (0.25° × 0.25°), PCA cluster analysis/Bani basin | ERA5 | Second-order polynomial | 11 | SST: NSE mean = 0.867 SST: NSE max = 0.926 SST: NSE min = 0.751 |
| In this study | Atlantic SST (0.25° × 0.25°), PCA cluster analysis/Senegal basin | ERA5 | Second-order polynomial | 11 | SST: NSE mean = 0.711 SST: NSE max = 0.916 SST: NSE min = 0.133 |
| Sittichok et al. [12,13] | Atlantic and Pacific SST/(2° × 2°), combination of regression, PCA and ACC/Sirba basin | Meteorology and Water Resource Centre of Ceara State, Brazil | Stepwise regression | 5 12 | Atlantic SST NSE = 0.231 Pacific SST NSE = 0.387 |
| Gado Djibo et al. [2] | SLP, RHUM, Ta, zonal wind and meridional wind/Sirba basin | NCEP- DOE Reanalysis (NOAA), 2.5° × 2.5° | Linear | SLP: 0 RHUM: 8 Ta: 7 VWIND: 8 UWIND: 7 SST: 12 | SLP: NSE = 0.46 RHUM: NSE = 0.52 Ta: NSE = 0.53 VWIND: NSE = 0.28 UWIND: NSE = 0.32 SST: NSE = 0.34 |
| | | | Non-linear | SLP: 9 RHUM: 7 Ta: 8 | SLP: NSE = 0.31 RHUM: NSE = 0.36 Ta: NSE = 0.45 |
| Gado Djibo et al. [31] | Ta, SLP, RHUM | Climatic Research Unit | Linear | Ta: 14 SLP: 0 RHUM: 8 | Ta: NSE = 0.76 SLP: NSE = 46 RHUM: NSE = 0.52 |
| Garric et al. [44] | Gulf of Guinea SST | CRU, NCEP/NCAR, ECMWF) | Linear, stepwise regression | 12 | SST: r = 0.67 |
| Folland et al. [4] | SST | Meteorological Office Historical Sea Surface Temperature data set version 3 (MOHSST3) | Stepwise regression Linear Discriminant | 1 | SST: r = 0.54 SST: r = 0.72 |

Furthermore, for the pixels of the Bani River basin, the minimum value of NSE was greater than 0.696, and the lowest maximum value was 0.869 for each of the four models. Those values for the pixels of the Senegal River basin were 0.164 and 0.855.

The monthly rainfall forecasts using the polynomial model of each pixel were calculated for the historical period of PERSIANN-CDR. Appendix A (Table A1) presents forecasts for the two basins. Comparison between forecasts and PERSIANN-CDR with their corresponding relative error (RE) are given in Appendix A (Table A2) at 20 pixels where gauges are located. RE is small for all pixels except for the one corresponding to Tidjikja, located in the Mauritanian desert. Overall mean relative error was about 12.6%.

Findings of research show that the SST of the north of the Tropical Atlantic and the SST of the Gulf of Guinea has a strong teleconnection with rainfall at the Sahel. The average NSE was 0.80, and the maximum NSE was 0.926. MAE was ~30 mm month$^{-1}$, and the polynomial model was the one with the lowest error. Gado Djibo et al. [31] obtained satisfactory results with the Bayesian method of multiple point change and air temperature (NSE = 0.76, lag = 14 months). In another study, Gado Djibo et al. [2] combined linear models and reported the following results, with air temperature (NSE = 0.53, lag 7 months), with SST (NSE = 0.34, lag = 12 months), RHUM (NSE = 0.52, lag = 8 months). It is worth noting that a non-linear model does not always turn out to be better than a linear model [31]. Results from the polynomial model are 26% and 42% better than those reported in previous studies [2,31] and 46% better than the model of [12] (Table 5).

## 5. Conclusions

The availability of long series of global satellite-based meteorological products with high spatial and temporal resolution is increasingly facilitating and stimulating the implementation of rainfall forecasting models, particularly in undergauged regions. More so, if one considers the challenges that arise in Africa due to climate change and intensifying rainfall variability. The forecasting models using these satellite datasets can provide



valuable up-to-date information useful to decision makers. Millions of people in the Sahel region, who have suffered from the effects of droughts and floods, will be able to benefit from the information provided by these forecasting tools to better anticipate the planning of activities related to water, particularly for agriculture. During the last decades, several authors tried to develop models that would predict whether the rainy season would be wet, normal, or dry in the Sahel region. Statistical seasonal rainfall forecasting models are more often used than physically based models because of their simplicity.

This research dealt with seasonal rainfall forecasting for the Bani and Senegal basins. The Atlantic sea surface temperature (SST), the mean sea level pressure (MSLP), the relative humidity (RHUM), and five El Niño indices were used as explanatory variables, and PERSIANN-CDR rainfall data were used as a dependent variable.

Lineal, polynomial, exponential, and stepwise regression models were developed to forecast rainfall for each of the 725 pixels of the two basins. These models were built using 70% of the satellite datasets available from 1983 to 2020. The remaining 30% was used for model validation. The strategy was to find a model that satisfied the criteria: R2adj > 0.5 and lower AIC value, as well as a model that has a lag (lead time) greater than or equal to six months, a reasonable time for authorities to take preventive measures.

Based on the principal component analysis and cluster analysis, three SST, three RHUM, and two MSLP homogeneous regions were defined. The study revealed that the North Atlantic SST, a region of approximately $11.5 \times 106$ km$^2$ (16°N–40°N y 9°W–39°W), has a better teleconnection with rainfall in the two basins with a lead time of eleven months. The Gulf of Guinea SST also has a suitable teleconnection with rainfall over the region but with a lead time of five months. Finally, all models were built using datasets of the North Atlantic region with a lead time of eleven months.

All four models provided suitable results in all the pixels of the Bani River basin based on the numerical criteria; the smallest NSE value was about 0.696 with the linear model.

For the pixels of the Senegal River basin, each of the four models provided suitable results, but up to latitude 16.5°N approximately. None of the models was able to forecast rainfall relatively precisely in the far north of the Senegal River basin that corresponds to the Sahara. For this region, other predictors (such as temperature, wind speed, and direction) or other types of models should be considered.

The results of the stepwise regression model are not very different from those of the linear model. This means that the other explanatory variables did not contribute much to explain the phenomenon. It is understandable because SST is the driving force of many of these variables.

The best of the four models for rainfall forecasting in the study area was the second-order polynomial model. For the Bani River basin, values of NSE were between 0.751 and 0.926, with a mean of about 0.867. However, for the Senegal River basin, these values were 0.133, 0.916, and 0.711, respectively. Moreover, the relative error calculated for the pixels where the rain gauges are located was globally quite low, sometimes even close to zero. Overall, it is easy to say that the polynomial model gives suitable rainfall forecasts.

The statistical models used in this study are easy to apply, and the satellite datasets are accessible to users. Finally, this study aims to make a significant contribution to improving the effectiveness of forecasts more than six months in advance, which is enough time for agricultural planning and decision making.

**Supplementary Materials:** A script for forecast models can be found at https://github.com/lebalcazar/sahel.

**Author Contributions:** Conceptualization, L.B., K.M.B. and C.D.-D.; methodology, L.B., K.M.B., C.D.-D., S.M.-L. and G.G.; software, L.B. and G.G.; validation, L.B., K.M.B. and C.D.-D.; formal analysis, L.B., S.M.-L. and G.G.; investigation, L.B., K.M.B., C.D.-D. and M.A.G.-A.; resources, L.B., K.M.B., C.D.-D. and M.A.G.-A.; data curation, L.B., S.M.-L. and G.G.; writing—original draft preparation, L.B., K.M.B., S.M.-L. and C.D.-D.; writing—review and editing, L.B., K.M.B., C.D.-D. and M.A.G.-A.; visualization, L.B. and G.G.; supervision, K.M.B., C.D.-D. and M.A.G.-A.; project administration,

K.M.B. and C.D.-D.; funding acquisition, K.M.B. and C.D.-D. All authors have read and agreed to the published version of the manuscript.

**Funding:** This research was funded by "La Fiducie pour la recherche en hydrologie, Québec" in memory of Late José Llamas and his wife Constance Gravel, grants [UAEM: 6314E/2021, 4192/2016].

**Data Availability Statement:** PERSIANN-CDR data are available at https://chrsdata.eng.uci.edu, accessed on 16 November 2022; SST data are available at https://cds.climate.copernicus.eu/cdsapp# !/dataset/reanalysis-era5-single-levels?tab=overview, accessed on 16 November 2022; MSPLP and RHUM data are available at https://cds.climate.copernicus.eu/cdsapp#!/dataset/reanalysis-era5 -single-levels-monthly-means?tab=overview, accessed on 16 November 2022; El Niño indices are available at https://climatedataguide.ucar.edu/climate-data/nino-sst-indices-nino-12-3-34-4-oni- and-tni, accessed on 16 November 2022; and PIRATA buoys data are available https://www.ndbc. noaa.gov/, accessed on 16 November 2022.

**Acknowledgments:** The authors would like to thank the OMVS and the National Meteorological and Hydrological Services of Cote d'Ivoire, Guinea, Mali, Mauritania, and Senegal for providing some meteorological data. This research has also been partially funded by the National Council for Science and Technology of Mexico (CONACyT) through a doctoral scholarship. Thanks to Mamoudou BA of NOAA and Alin CARSTEANU of IPN for their suggestions. In addition, thanks to the four anonymous reviewers for their contributions.

**Conflicts of Interest:** The authors declare no conflict of interest.

## Appendix A

**Table A1.** Monthly rainfall forecasts (mm) in the Bani and Senegal River basins.

| Year | Jan. | Feb. | Mar. | Apr. | May | Jun. | Jul. | Aug. | Sep. | Oct. | Nov. | Dec. | Total |
|------|------|------|------|------|------|------|------|------|------|------|------|------|-------|
| 1983 | - | - | - | - | - | - | - | - | - | - | - | 0 | - |
| 1984 | 0 | 0 | 0 | 0 | 26.1 | 82.4 | 133.6 | 152 | 96.7 | 35.4 | 0 | 0 | 526 |
| 1985 | 0 | 0 | 0 | 0 | 19 | 78.2 | 127.7 | 140.8 | 92.1 | 34.6 | 0 | 0 | 492 |
| 1986 | 0 | 0 | 0 | 0 | 38.1 | 106.3 | 153.8 | 157.1 | 90.9 | 29.3 | 0 | 0 | 576 |
| 1987 | 0 | 0 | 0 | 0 | 33.5 | 98.8 | 159 | 135.2 | 82.7 | 36.6 | 0 | 0 | 546 |
| 1988 | 0 | 0 | 0 | 0 | 37.6 | 117 | 162.5 | 174.1 | 108.6 | 37.1 | 0 | 0 | 637 |
| 1989 | 0 | 0 | 0 | 0 | 40.9 | 101.8 | 174.6 | 162.1 | 107.1 | 37.9 | 0 | 0 | 624 |
| 1990 | 0 | 0 | 0 | 0 | 57.1 | 131.2 | 189.3 | 158.9 | 95.5 | 36.1 | 0 | 0 | 668 |
| 1991 | 0 | 0 | 0 | 0 | 36.9 | 126 | 182.5 | 194.7 | 117.8 | 48.4 | 11.7 | 0 | 718 |
| 1992 | 0 | 0 | 0 | 0 | 37.9 | 115.8 | 195.4 | 182.9 | 100.9 | 42 | 10.6 | 0 | 686 |
| 1993 | 0 | 0 | 0 | 0 | 33.8 | 98.7 | 170.2 | 160.2 | 89.1 | 30.7 | 10.2 | 0 | 593 |
| 1994 | 0 | 0 | 0 | 0 | 31.6 | 85.6 | 154.1 | 161 | 93.7 | 35.5 | 0 | 0 | 562 |
| 1995 | 0 | 0 | 0 | 0 | 37.2 | 111.6 | 166.7 | 167.6 | 108.4 | 51.9 | 13.3 | 0 | 657 |
| 1996 | 0 | 0 | 0 | 10.4 | 54.8 | 133.8 | 186.5 | 170.2 | 111.9 | 44.9 | 0 | 0 | 712 |
| 1997 | 0 | 0 | 0 | 0 | 29.5 | 91 | 163.9 | 171.1 | 111.7 | 46.6 | 0 | 0 | 614 |
| 1998 | 0 | 0 | 0 | 0 | 34.2 | 100.9 | 171.8 | 173.1 | 111.7 | 45.2 | 10.9 | 0 | 648 |
| 1999 | 0 | 0 | 0 | 0 | 47.5 | 118.8 | 184.1 | 186.6 | 125.5 | 71.4 | 19.1 | 0 | 753 |
| 2000 | 0 | 0 | 0 | 10.5 | 42.7 | 129.6 | 201.4 | 195.1 | 122.2 | 47.8 | 11.4 | 0 | 761 |
| 2001 | 0 | 0 | 0 | 0 | 45.2 | 111.8 | 165.4 | 171.3 | 118.1 | 51.3 | 11.3 | 0 | 674 |
| 2002 | 0 | 0 | 0 | 0 | 40.2 | 117.1 | 198.7 | 196.2 | 135.2 | 52 | 13 | 0 | 752 |
| 2003 | 0 | 0 | 0 | 0 | 34.2 | 92.9 | 160.8 | 165.6 | 110.3 | 48.4 | 13.2 | 0 | 625 |
| 2004 | 0 | 0 | 0 | 0 | 54.4 | 151.5 | 219.2 | 203.5 | 126.5 | 52.3 | 14.5 | 0 | 822 |
| 2005 | 0 | 0 | 0 | 0 | 54.7 | 132.6 | 207.5 | 186.1 | 122.3 | 54.5 | 15.3 | 0 | 773 |
| 2006 | 0 | 0 | 0 | 0 | 38.3 | 122.4 | 180.3 | 177.5 | 114.7 | 51 | 14.2 | 0 | 698 |
| 2007 | 0 | 0 | 0 | 0 | 43.6 | 120.9 | 194.4 | 182 | 121.7 | 59.8 | 14.7 | 0 | 737 |
| 2008 | 0 | 0 | 0 | 0 | 33.7 | 102.3 | 152.8 | 152.6 | 111.1 | 50.6 | 13.6 | 0 | 617 |
| 2009 | 0 | 0 | 0 | 0 | 53 | 129.9 | 188.5 | 190.1 | 116.8 | 45.5 | 10.3 | 0 | 734 |
| 2010 | 0 | 0 | 0 | 0 | 48.9 | 132.1 | 202.8 | 190.3 | 130.2 | 58.3 | 14.1 | 0 | 777 |
| 2011 | 0 | 0 | 0 | 0 | 37.6 | 121 | 184.8 | 179.4 | 124.8 | 57.9 | 12.1 | 0 | 718 |
| 2012 | 0 | 0 | 0 | 0 | 41.4 | 107.2 | 165.6 | 190.6 | 126.8 | 47.1 | 12.9 | 0 | 692 |
| 2013 | 0 | 0 | 0 | 0 | 48.1 | 117.6 | 211.2 | 198.3 | 121.3 | 47.8 | 11.7 | 0 | 756 |
| 2014 | 0 | 0 | 0 | 0 | 37.8 | 126.7 | 182.1 | 175.7 | 125.2 | 53.4 | 12.5 | 0 | 713 |
| 2015 | 0 | 0 | 0 | 0 | 41.1 | 126.1 | 190.6 | 200.2 | 134.7 | 60.2 | 13.4 | 0 | 766 |
| 2016 | 0 | 0 | 0 | 0 | 52.5 | 148.2 | 207.4 | 211.9 | 136.1 | 55.2 | 15.8 | 0 | 827 |
| 2017 | 0 | 0 | 0 | 11.5 | 52.2 | 121.8 | 187.3 | 183.4 | 123.7 | 58.5 | 16.2 | 0 | 755 |
| 2018 | 0 | 0 | 0 | 0 | 49.6 | 126.6 | 193 | 183.1 | 134.1 | 68.7 | 21.9 | 0 | 777 |
| 2019 | 0 | 0 | 0 | 10.1 | 47.6 | 111.9 | 178.9 | 194.1 | 121.2 | 48.6 | 14.2 | 0 | 727 |
| 2020 | 0 | 0 | 0 | 0 | 48.7 | 125.7 | 209.3 | 196.1 | 121.7 | 56.7 | 17.9 | 0 | 776 |

**Table A2.** Validation of the forecasts (FRC) with PERSIANN-CDR rainfall (CDR) at pixel-to-point from the last 5 years (complement of Table 4).

| Name | Year | CDR | FRC | ER | Name | Year | CDR | FRC | ER |
|---|---|---|---|---|---|---|---|---|---|
| Bakel | 2017 | 519 | 613 | 18.1 | Korhogo | 2017 | 846 | 996 | 17.7 |
| Bakel | 2018 | 596 | 639 | 7.2 | Korhogo | 2018 | 1043 | 1013 | 2.9 |
| Bakel | 2019 | 530 | 591 | 11.5 | Korhogo | 2019 | 1141 | 976 | 14.5 |
| Bakel | 2020 | 722 | 643 | 10.9 | Korhogo | 2020 | 935 | 1008 | 7.8 |
| Bakel | 2021 | 581 | 653 | 12.4 | Korhogo | 2021 | 904 | 1016 | 12.4 |
| Bamako-Senou | 2017 | 950 | 1057 | 11.3 | Mamou | 2017 | 1801 | 1734 | 3.7 |
| Bamako-Senou | 2018 | 1015 | 1096 | 8.0 | Mamou | 2018 | 1497 | 1764 | 17.8 |
| Bamako-Senou | 2019 | 1318 | 1022 | 22.5 | Mamou | 2019 | 1743 | 1698 | 2.6 |
| Bamako-Senou | 2020 | 1219 | 1100 | 9.8 | Mamou | 2020 | 1624 | 1753 | 7.9 |
| Bamako-Senou | 2021 | 1246 | 1116 | 10.4 | Mamou | 2021 | 1692 | 1765 | 4.3 |
| Bamako-Ville | 2017 | 908 | 1011 | 11.3 | Matam | 2017 | 298 | 354 | 18.8 |
| Bamako-Ville | 2018 | 963 | 1050 | 9.0 | Matam | 2018 | 303 | 370 | 22.1 |
| Bamako-Ville | 2019 | 1236 | 977 | 21.0 | Matam | 2019 | 260 | 340 | 30.8 |
| Bamako-Ville | 2020 | 1184 | 1053 | 11.1 | Matam | 2020 | 492 | 373 | 24.2 |
| Bamako-Ville | 2021 | 1150 | 1069 | 7.0 | Matam | 2021 | 425 | 380 | 10.6 |
| Goudiry | 2017 | 717 | 781 | 8.9 | Odienne | 2017 | 1230 | 1365 | 11.0 |
| Goudiry | 2018 | 730 | 813 | 11.4 | Odienne | 2018 | 1395 | 1397 | 0.1 |
| Goudiry | 2019 | 686 | 754 | 9.9 | Odienne | 2019 | 1668 | 1332 | 20.1 |
| Goudiry | 2020 | 755 | 817 | 8.2 | Odienne | 2020 | 1463 | 1394 | 4.7 |
| Goudiry | 2021 | 780 | 830 | 6.4 | Odienne | 2021 | 1441 | 1407 | 2.4 |
| Kayes | 2017 | 780 | 815 | 4.5 | San | 2017 | 742 | 790 | 6.5 |
| Kayes | 2018 | 794 | 851 | 7.2 | San | 2018 | 855 | 824 | 3.6 |
| Kayes | 2019 | 797 | 784 | 1.6 | San | 2019 | 826 | 761 | 7.9 |
| Kayes | 2020 | 896 | 857 | 4.4 | San | 2020 | 917 | 829 | 9.6 |
| Kayes | 2021 | 834 | 871 | 4.4 | San | 2021 | 722 | 843 | 16.8 |
| Kedougou | 2017 | 1223 | 1261 | 3.1 | Siguiri | 2017 | 1068 | 1282 | 20.0 |
| Kedougou | 2018 | 1112 | 1307 | 17.5 | Siguiri | 2018 | 1193 | 1321 | 10.7 |
| Kedougou | 2019 | 1254 | 1220 | 2.7 | Siguiri | 2019 | 1291 | 1245 | 3.6 |
| Kedougou | 2020 | 1220 | 1310 | 7.4 | Siguiri | 2020 | 1277 | 1321 | 3.4 |
| Kedougou | 2021 | 1210 | 1329 | 9.8 | Siguiri | 2021 | 1413 | 1337 | 5.4 |
| Kiffa | 2017 | 343 | 321 | 6.4 | Sikasso | 2017 | 988 | 1105 | 11.8 |
| Kiffa | 2018 | 328 | 336 | 2.4 | Sikasso | 2018 | 1252 | 1141 | 8.9 |
| Kiffa | 2019 | 265 | 309 | 16.6 | Sikasso | 2019 | 1168 | 1072 | 8.2 |
| Kiffa | 2020 | 444 | 339 | 23.6 | Sikasso | 2020 | 1172 | 1142 | 2.6 |
| Kiffa | 2021 | 308 | 345 | 12.0 | Sikasso | 2021 | 1123 | 1157 | 3.0 |
| Kita | 2017 | 1024 | 1138 | 11.1 | Tidjikja | 2017 | 103 | 133 | 29.1 |
| Kita | 2018 | 1116 | 1182 | 5.9 | Tidjikja | 2018 | 106 | 140 | 32.1 |
| Kita | 2019 | 1288 | 1100 | 14.6 | Tidjikja | 2019 | 74 | 128 | 73.0 |
| Kita | 2020 | 1352 | 1186 | 12.3 | Tidjikja | 2020 | 182 | 141 | 22.5 |
| Kita | 2021 | 1219 | 1204 | 1.2 | Tidjikja | 2021 | 60 | 143 | 138.3 |

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
