# Peer review of "Development and Assessment of Seasonal Rainfall Forecasting Models for the Bani and the Senegal Basins by Identifying the Best Predictive Teleconnection"

_remotesensing, doi:10.3390/rs14246397_

Round 1

Reviewer 1 Report

Comments for remotesensing-2008710

General comments:

This study constructed four seasonal rainfall forecasting model and assessed them at Bani and Senegal basins. The conclusions is not rigorous, and some results do not seem relevant to the article. Overall, I suggestion a major revision.

Specific comments:

In abstract: the sentences “All three models use Atlantic Sea Surface Temperature (SST).” “The stepwise regression algorithm uses SST with covariates, Mean Sea Level Pressure 21 (MSLP), Relative Humidity (RHUM) and five El Niño indices.” “ PERSIANN-CDR rainfall data were used as dependent variable.” are not completely. “best” in last sentence is not accurate, it should be “better”.

Keywords: “Sahel” did not appear in abstract, why is it a keyword? In addition, “model” should be not a keyword.

Introduction: The time is not accurate, for example, 1970s should be from 1970 to 1979, not from 1972 to 1974; as well as 1980s.

Results: It is not reasonable to compare the PERSIANN-CDR with rain gauge observations directly, because they are in two large different scales, i.e., point and 0.25 degree, even though the comparable results seems good, which may be caused by the coarse temporal scale (month). In addition, it does not seem relevant to the article.

Table 2 lacks the bottom line.

Conclusion: “Results of this work revealed that the second-order polynomial model with the 492 North Atlantic SST is the best predictor of rainfall for the Bani and Senegal River basins 493 with a lag of 11 months.” is not reasonable, because the authors only use four equations, maybe other model without used here is better than these four models. In addition, whether the lag time “11 months” is too long? I don’t think the too long lag time of the model is a good model.

The English language should be improved.

Author Response

General comments:

This study constructed four seasonal rainfall forecasting model and assessed them at Bani and Senegal basins. The conclusions is not rigorous, and some results do not seem relevant to the article. Overall, I suggestion a major revision.

Specific comments:

In abstract: the sentences “All three models use Atlantic Sea Surface Temperature (SST).” “The stepwise regression algorithm uses SST with covariates, Mean Sea Level Pressure 21 (MSLP), Relative Humidity (RHUM) and five El Niño indices.” “ PERSIANN-CDR rainfall data were used as dependent variable.” are not completely. “best” in last sentence is not accurate, it should be “better”.

Change has been made: …SST shows a more robust teleconnection …. See lines 33-34.

Keywords: “Sahel” did not appear in abstract, why is it a keyword? In addition, “model” should be not a keyword.

Agreed: Sentence has been added. Please see line 18.

Introduction: The time is not accurate, for example, 1970s should be from 1970 to 1979, not from 1972 to 1974; as well as 1980s.

Yes that is correct but we wanted to emphasize that during the these years drought was severe.

Changes have been made.

See: Line 42

Results: It is not reasonable to compare the PERSIANN-CDR with rain gauge observations directly, because they are in two large different scales, i.e., point and 0.25 degree, even though the comparable results seems good, which may be caused by the coarse temporal scale (month). In addition, it does not seem relevant to the article.

PERSIANN-CDR are not gauged data, the least we can do is show that the monthly data are comparable. Please see references:  [8], [25] and [26].

Table 2 lacks the bottom line.

Agreed: change has been made lines 315 to 316.

 Conclusion: “Results of this work revealed that the second-order polynomial model with the North Atlantic SST is the best predictor of rainfall for the Bani and Senegal River basins with a lag of 11 months.” is not reasonable, because the authors only use four equations, maybe other model without used here is better than these four models. In addition, whether the lag time “11 months” is too long? I don’t think the too long lag time of the model is a good model.

Changes have been made please see lines 539-544

The English language should be improved.

Reviewer 2 Report

I read the manuscript with interest. The subject of forecasts is very necessary and important, especially for areas suffering from water deficits. However, I have doubts as to the applicability of the obtained results.

This was also demonstrated by the results of the study in the northern part of the Senegal River basin (desert area). I am asking you to expand on this issue in the discussion. What could be the reasons? Can this be avoided?

I also have doubts what was the guiding principle behind the selection of these and not other catchments? How were the catchment boundaries determined (whole areas were not covered, only fragments). This creates some confusion in the interpretation of the results.

The methodology itself is acceptable, but haven't you been addressing an area that is too difficult, where more factors affect the weather conditions?

Author Response

English language and style

( ) English very difficult to understand/incomprehensible
( ) Extensive editing of English language and style required
(x) Moderate English changes required
( ) English language and style are fine/minor spell check required
( ) I don't feel qualified to judge about the English language and style

Yes

Can be improved

Must be improved

Not applicable

( )

(x)

( )

( )

( )

(x)

( )

( )

( )

(x)

( )

( )

( )

(x)

( )

( )

( )

(x)

( )

( )

( )

(x)

( )

( )

Comments and Suggestions for Authors

I read the manuscript with interest. The subject of forecasts is very necessary and important, especially for areas suffering from water deficits. However, I have doubts as to the applicability of the obtained results.

Please see the following 5 references 12, 13, 2, 31, 4

Autor

Criterios utilizados en evaluación

In this study

NSE*, R2, R2adj, AIC, NSE, PBIAS, RE, MAE

Sittichok (2015)

NSE, R2, HIT-score,

Sittichok (2016)

NSE, R2, HIT-score,

Gado (2015) Climate

NSE, R2, HIT-score

Gado (2015) J.Hyd. Regional Studies

NSE, R2, HIT-score

Folland (1991)

R, ABSE, RMSE, BIAS, sd

This was also demonstrated by the results of the study in the northern part of the Senegal River basin (desert area). I am asking you to expand on this issue in the discussion. What could be the reasons? Can this be avoided?

I also have doubts what was the guiding principle behind the selection of these and not other catchments? How were the catchment boundaries determined (whole areas were not covered, only fragments). This creates some confusion in the interpretation of the results.

The final objective is to simulate runoff of these rivers. I did previously simulations using gauged rainfall and PERSIANN-CDR. Now, following this paper, we plane to use the forecasted rainfall for runoff forecasting. Daily flows are available at Bakel (Senegal River and at Beneny Kegny (Bani River) gauges. Each basin boundary has been defined using SRTM images of USGS with the help of GIS software.

The methodology itself is acceptable, but haven't you been addressing an area that is too difficult, where more factors affect the weather conditions?

You are right, it is a complex region from a climatic dynamic point of view. Several complex physical factors are in play, that is why it seems reasonable to explore statistical methods as several authors have done

Reviewer 3 Report

This study aims to establish a forecasting model of rainfall during May to October for Bani and Senegal River basins. Overall, I think this manuscript is well written and organized, so I recommend minor revision for now, some comments are listed below for your reference.

1.     Figure 6 looks confused, can author give more description about the meaning of each histogram, and what’s the meaning of each color, and what’s the meaning of each color’s height and how did they get it?

2.     Figure 7, why does the over normal occupy the majority?

3.     Discussion section may need more description about the uncertainties of this research and what’s the future development or implication for other studies.

Author Response

Comments and Suggestions for Authors

This study aims to establish a forecasting model of rainfall during May to October for Bani and Senegal River basins. Overall, I think this manuscript is well written and organized, so I recommend minor revision for now, some comments are listed below for your reference.

  1. Figure 6 looks confused, can author give more description about the meaning of each histogram, and what’s the meaning of each color, and what’s the meaning of each color’s height and how did they get it?

This is explained between lines 389 and 394. The methodology is similar to the Flow Duration curve procedure, but here since the variable is precipitation, we have represented it as hyetograph.

  1. Figure 7, why does the over normal occupy the majority?

Because 2021 has been a wet year according to PERSIANN-CDR data and to forecasts.

If rainfall is greater than quantile corresponding to a probability of 0.70 then it is considered as wet. Please see lines 389 to 394.

3.Discussion section may need more description about the uncertainties of this research

Agreed, changes have been made: please see Lines 466-480.

Reviewer 4 Report

REVIEW of the manuscript "Development and assessment of seasonal rainfall forecasting models for the Bani and the Senegal basins by identifying the best predictive teleconnection" by Luis Balcázar, Khalidou M. Bâ, Carlos Díaz-Delgado, Miguel A. Gómez-Albores, Gabrie Gaona and Saula Minga-León [Remote Sens. 2022, 14, x. https://doi.org/10.3390/xxxxx].

Due to the higher accuracy of the rainfall forecast, the authors of this study have developed linear, polynomial and exponential models to forecast rainfall in the Bani and Senegal River basins that use Atlantic Sea Surface Temperature (SST). The authors also use a fourth algorithm with stepwise regression using SST with covariates, Mean Sea Level Pressure  (MSLP), Relative Humidity (RHUM) and five El Niño indices. The first three variables were selected based on principal component analysis (PCA) and cluster analysis. PERSIANN-CDR rainfall data were used as dependent variable. The spatial resolution of proposed models was for each pixel of 0.25° x 0.25°. Statistical indicators showed that the north Atlantic SST has the best teleconnection with precipitation dynamics in both basins. The weakest results are for the driest area.

The models presented in this study represent progress in the rainfall forecast for the observed area and give promising results. The authors have explained in detail what is new in their research as well as the significance and objectives of the study. The usefulness of the study results in the form of a recommended statistical model for rainfall forecast is also clearly visible. However, this study should be complemented by a physical essence of the relationships between the processes and parameters (e.g. teleconnection) used in the statistical model (for example included in the analysis of the results). I would therefore suggest publication of this paper after the proposed addition.

Author Response

REVIEW of the manuscript "Development and assessment of seasonal rainfall forecasting models for the Bani and the Senegal basins by identifying the best predictive teleconnection" by Luis Balcázar, Khalidou M. Bâ, Carlos Díaz-Delgado, Miguel A. Gómez-Albores, Gabrie Gaona and Saula Minga-León [Remote Sens. 2022, 14, x. https://doi.org/10.3390/xxxxx].

Due to the higher accuracy of the rainfall forecast, the authors of this study have developed linear, polynomial and exponential models to forecast rainfall in the Bani and Senegal River basins that use Atlantic Sea Surface Temperature (SST). The authors also use a fourth algorithm with stepwise regression using SST with covariates, Mean Sea Level Pressure  (MSLP), Relative Humidity (RHUM) and five El Niño indices. The first three variables were selected based on principal component analysis (PCA) and cluster analysis. PERSIANN-CDR rainfall data were used as dependent variable. The spatial resolution of proposed models was for each pixel of 0.25° x 0.25°. Statistical indicators showed that the north Atlantic SST has the best teleconnection with precipitation dynamics in both basins. The weakest results are for the driest area.

The models presented in this study represent progress in the rainfall forecast for the observed area and give promising results. The authors have explained in detail what is new in their research as well as the significance and objectives of the study. The usefulness of the study results in the form of a recommended statistical model for rainfall forecast is also clearly visible. However, this study should be complemented by a physical essence of the relationships between the processes and parameters (e.g. teleconnection) used in the statistical model (for example included in the analysis of the results). I would therefore suggest publication of this paper after the proposed addition.

Agreed: please see Lines 466 to 471.

Round 2

Reviewer 1 Report

The authors did not answer some comments before directly, and I don't think some others responses are acceptable. So, I suggest to reject it.

Author Response

Thank you for your relevant comments. We think that we have taken into account all your comments. We have made changes to the abstract, to the discussions and have reworded the conclusions. English language has been improved.

Reviewer 4 Report

REVIEW of the revised version of the manuscript "Development and assessment of seasonal rainfall forecasting models for the Bani and the Senegal basins by identifying the best predictive teleconnection" by Luis Balcázar, Khalidou M. Bâ, Carlos Díaz-Delgado, Miguel A. Gómez-Albores, Gabrie Gaona and Saula Minga-León [Remote Sens. 2022, 14, x. https://doi.org/10.3390/xxxxx].

The revised version of the manuscript is essentially improved compared to the original version. The present version of the manuscript partly takes into account my principal comment. In my opinion, this is sufficient that this manuscript would be acceptable for publishing in this eminent journal.

Author Response

Dear reviewer,

Thank you for your relevant comments. We think that we have taken into account all your comments. We have made changes to the abstract, to the discussions and have reworded the conclusions. Also English language has been improved.